# Seeing and Sensing the Hepatorenal Syndrome (HRS): The Growing Role of Ultrasound-Based Techniques as Non-Invasive Tools for the Diagnosis of HRS

**DOI:** 10.3390/diagnostics14090938

**Published:** 2024-04-30

**Authors:** Cornelia Tăluță, Horia Ștefănescu, Dana Crișan

**Affiliations:** 1Liver Unit, Regional Institute of Gastroenterology and Hepatology, 400162 Cluj-Napoca, Romania; corneliataluta@gmail.com; 25th Medical Clinic, Department of Internal Medicine, Iuliu Hatieganu University of Medicine and Pharmacy, 400139 Cluj-Napoca, Romania; crisan.dc@gmail.com

**Keywords:** cirrhosis, acute kidney injury, hepatorenal syndrome, urinary biomarkers, shear wave elastography

## Abstract

More than half of patients hospitalized with liver cirrhosis are dealing with an episode of acute kidney injury; the most severe pattern is hepatorenal syndrome due to its negative prognosis. The main physiopathology mechanisms involve renal vasoconstriction and systemic inflammation. During the last decade, the definition of hepatorenal syndrome changed, but the validated criteria of diagnosis are still based on the serum creatinine level, which is a biomarker with multiple limitations. This is the reason why novel serum and urinary biomarkers have been intensively studied in recent years. Meanwhile, the imaging studies that use shear wave elastography are using renal stiffness as a surrogate for an early diagnosis. In this article, we focus on the physiopathology definition and highlight the novel tools used in the diagnosis of hepatorenal syndrome.

## 1. Introduction: The Spectrum of Acute Kidney Injuries in Cirrhosis

Acute kidney injury (AKI) is a common complication occurring in patients with advanced liver disease and is clinically manifested as prerenal AKI, hepatorenal syndrome AKI, intrarenal AKI—which is mainly represented by acute tubular necrosis—or postrenal AKI (obstructive uropathies) [1].

Almost 68% of patients hospitalized with liver cirrhosis are dealing with prerenal AKI; most cases are resolved by fluid expansion with albumin. Nevertheless, more than 25% of renal injury cases develop before hospitalization; this is known as community-acquired AKI. For this group, it is mandatory to have a reference serum creatinine level before admission to differentiate between chronic kidney disease and a new onset of renal injury [1]. There are three stages of AKI; these stages depend on the value of the serum creatinine level (sCr), as we discuss in the following sections. The progression of AKI through the stages (stages 1, 2, or 3) is strongly correlated with an increased mortality rate [2].

In recent years, there have been several important changes in both the classification and the pathophysiology of AKI, with the identification of new connections between chronic liver disease and renal dysfunction. According to the new theory, the hemodynamic changes, which are due to renal vasoconstriction, are not the only mechanism responsible for the development of hepatorenal syndrome. The hypothesis that systemic inflammation with an increase in pro-inflammatory cytokines may have a role in the development of HRS was intensively studied [3,4].

The nomenclature was also changed. HRS is currently defined as a phenotype of renal injury that occurs in patients with advanced liver disease, particularly in those decompensated with ascites; it is often precipitated by a bacterial infection and sometimes by gastrointestinal bleeding and large volume paracentesis without albumin administration, or it is secondary to diuretics, or other hepatic factors (alcohol abuse or hepatitis flares). The sub-types of HRS have been changed based on whether the renal failure is acute (AKI), sub-acute (AKD), or chronic (CKD) [5].

One of the main problems of renal injury in liver cirrhotic patients is the differentiation between HRS-AKI and ATN. While HRS is partially considered to be a functional injury, kidney biopsy is rarely performed for a confirmation of ATN. Consequently, novel biomarkers are being studied to improve the diagnosis and initiation of early treatment; these biomarkers include urinary neutrophil gelatinase lipocalin (NGAL), Cystatin (Cys), kidney injury molecule 1 (KIM-1), fatty acid-binding proteins (FABPs), interleukin-18 (Il-18), or urine N-acetyl-β-D-glucosaminidase (NAG), but they still need further validation [6]. A biomarker of futility during HRS therapy is lacking. Point-of-care ultrasound (POCUS) is a helpful tool to exclude intrinsic (especially obstructive) renal conditions. Dynamic ultrasound techniques (such as pulsed Doppler or contrast-enhanced ultrasound (CEUS)) showed promising results in some studies. Renal tissue stiffness assessed by real-time shear wave elastography (SWE) was also evaluated in patients with kidney disease or in experimental settings, but to a lesser extent in the context of HRS. However, renal SWE seems to be a promising and helpful biomarker for the diagnosis and prognosis of this condition.

This review summarizes the existing data on HRS biomarkers—both serum or urinary and physical (ultrasound-based)—and explores further developments in their use, with an emphasis on renal SWE.

## 2. Hepatorenal Syndrome

### 2.1. General Approach

Hepatorenal syndrome is one of the most important complications in the natural history of decompensated advanced liver disease because of the associated high mortality rate and poor prognosis; it occurs in nearly 50% of patients hospitalized with cirrhosis [5].

Prior to the initiation of the specific treatment with vasoactive drugs, the algorithm of the management of AKI requires that all medication taken by the patient be reviewed, with the withdrawal of all potential nephrotoxic drugs, including diuretics, non-selective betablockers, or non-steroidal anti-inflammatory drugs (NSAIDs); early recognition and treatment of bacterial infections, as well as plasma volume expansion with albumin, are also required [7].

The actual standard of care for HRS involves the combination of volume expansion (albumin 20–40 g daily) with a vasoconstrictor agent (Terlipressin is the first choice, but norepinephrine or midodrine plus octreotide can be used) [1]. The mechanism through which vasoconstrictive drugs counteract hypovolemia due to splanchnic vasodilation is paradoxical and poorly understood; however, these drugs eventually improve renal perfusion [8]. According to EASL guidelines, a complete response is defined as a decrease in sCr from its peak to a value not higher than 0.3 mg/dL above the baseline, while the partial response is defined as a regression of sCr to a value which is higher than 0.3 mg/dL above the baseline. Terlipressin should be titrated gradually up to maximum of 12 mg per day and should be stopped after 14 days if there is a lack of response [7]. The most feared side effects of Terlipressin are cardiovascular ischemic events; because of these events, the discontinuation rate of the treatment is around 20%. In order to reduce the adverse effects, the continuous intravenous infusion of Terlipressin could be an alternative option [9]. However, adverse events related to Terlipressin may occur, even when using this route; in this case, its administration should be discontinued. Other situations in which the early discontinuation of Terlipressin is recommended, according to the most recent consensus on HRS issued by the Acute Disease Quality Initiative (ADQI) and International Club of Ascites (ICA), are: (a) SCr returns within 0.3 mg/dL of baseline, (b) kidney function does not improve after 48 h on maximum tolerated doses, and (c) RRT is indicated [10]. The algorithm for the management of AKI in the cirrhotic group of patients, with a focus on HRS, is resumed in Figure 1.

There are few studies assessing response predictors in HRS. The negative predictors are median arterial pressure (MAP) without an increase, with 5 mmHg at day 3 of the treatment, and advanced liver dysfunction (total bilirubin level > 10 mg/dL), as demonstrated by one study [11]. Another study showed better prognosis in patients with systemic inflammatory response syndrome (SIRS), with a rate of response to Terlipressin of 42% vs. 6.2% [12]. Nevertheless, it must be noted that no futility rule has been implemented to stop vasoactive therapy earlier in the event of a negative outcome and, consequently, to prevent further complications related to vasopressors.

Around 40–50% of patients diagnosed with HRS achieve a favorable outcome; however, the rate of recurrence is relatively high, around 30%. For this reason, the treatment mentioned above is considered to be a bridging therapy to liver transplantation [13].

### 2.2. Pathophysiology of Hepatorenal Syndrome

One of the main characteristics of advanced liver disease is splanchnic arterial vasodilation. This leads to a decrease in effective arterial blood volume and, consequently, to reduced systemic vascular resistance and increased cardiac output, as an adaptative compensatory mechanism [14]. In the decompensated phase of the disease, this balance is no longer efficient because of the increased synthesis of vasodilatation molecules, of which nitric oxide is the most well-known [15]. Secondary to effective hypovolemia, the vasoconstriction in the renal arterial vessels respond with the activation of the systemic vasoconstrictor mechanisms (the renin–angiotensin–aldosterone system, the increase in the production of vasoconstrictor agents—vasopressin—and the sympathetic nervous system), helping to preserve the renal flow in the first phase. These compensatory mechanisms are no longer effective if the liver disease progresses; consequently, the renal vasoconstriction leads to a decreased glomerular filtration rate, increased water retention, which leads to edema and ascites, hyponatremia due to solute-free water elimination, and, eventually, to the installation of hepatorenal syndrome [16].

In the last decade, the leading role that systemic inflammation plays in the production mechanism of acute kidney injuries in cirrhotic patients has been well recognized; frequently, these injuries are due to pathological bacterial translocation or overt bacterial infections by pathogen-associated molecular patterns (PAMPs), like endotoxins or bacterial DNA [16]. This leads to monocyte activation, which results in the release of proinflammatory cytokines (tumor necrosis factor alpha, interleukin 6, or interleukin 1 beta) [17]. The bacterial translocation is also responsible for multiple gene encoding molecules [18], of which toll-like receptor 4 (TLR 4) is the one mainly studied [19]. The upregulation of this factor in cirrhotic patients, at the renal level, was associated with kidney injury with morphological changes, such as tubular cell damage or apoptosis.

As summarized in Figure 2, the physiopathology of hepatorenal syndrome involves complex, co-dependent mechanisms, namely hemodynamic microvascular changes in the kidney and inflammatory response—which are frequently due to bacterial translocation. This pattern could explain why there is a category of patients who do not respond to the conventional hepatorenal syndrome treatment [20].

## 3. Diagnosis Tools

Concerning the diagnosis of AKI and the sub-types, including hepatorenal syndrome, the validated criteria are still based on the serum creatinine level. The advantages of a simple operation and low cost still make the sCr the preferred diagnosis tool in clinical practice [7].

The current definition, according to the International Club of Ascites, takes into account the dynamic changes in the serum creatinine level (sCr), with an increase of 0.3 mg/dL within 48 h or of ≥50% from the baseline, together with a reduction in urine output to <0.5 mL/kg/h for >6 h, as can be seen in Table 1. Those actual diagnostic criteria were recommended by a panel of experts from the Kidney Disease: Improving Global Outcomes (KDIGO) group, who suggested combining part of the Acute Kidney Injury Network (AKIN) criteria with the Acute Dialysis Quality Initiative group for the Risk, Injury, Failure, Loss of Renal Function and End-Stage Renal Disease (RIFLE) criteria. The first group proposed an increase in sCr of 0.3 mg/dL within 48 h or of ≥50% from the baseline and the reduction in urine output to less than 0.5 mL/kg/h for more than 6 h [21]. The RIFLE group suggested using an increase in sCr of more than 50% within 1 week or a reduction in GFR of more than 25% and a reduction in urine production to less than 0.5 mL/kg/h for more than 6 h [22].

Regarding the urine output in patients with decompensated advanced chronic liver disease, an important issue is represented by oliguria due to sodium retention, but it still has a relatively normal glomerular filtration rate (GFR) per 24 h [23]. Diuretic utilization and inaccurate urine collection also make urine output an imprecise means of predicting AKI [7].

The previous criteria were based on the threshold value of sCr > 1.5 mg/dL. Two main significant problems arose from that; firstly, this threshold could signify a markedly decreased glomerular filtration rate (GFR); the second issue is that this value does not count the dynamic changes in sCr in the preceding weeks and does not show the difference between an acute and a chronic onset of kidney injury [7].

However, sCr has some limitations that cannot be neglected, especially in a patient with advanced liver disease [24]. First of all, the level of sCr is influenced by body weight, race, and age. Secondly, in the cirrhotic population, the diagnosis value could be affected by the decreased formation of creatinine from creatine in muscles because of sarcopenia [25], decreased hepatic synthesis, increased volume of distribution in patients with ascites, interferences with elevated serum bilirubin level [26], and increased renal tubular secretion of creatinine [27].

According to the International Club of Ascites, a management algorithm for patients with AKI and cirrhosis was proposed based on the new diagnosis criteria. In order to accelerate the differential diagnosis process between prerenal AKI and the other types of AKI, according to the algorithm, the withdrawal of diuretics and other nephrotoxic drugs should be implemented and an ultrasonography to exclude an obstructive AKI should be performed, as should the plasma volume expansion, for at least 48 h, as specified in Table 2 [7]. According to the latest ADQI and IAC joint consensus, plasma expansion with albumin alone (1 g per kg bodyweight per day) is not recommended anymore for this purpose, except where it is specifically indicated [10]. It must be emphasized that the diagnosis of HRS remains an exclusionary one in patients with decompensated advanced liver disease and AKI.

For that reason, the delay between the altered renal function and the increase in the sCr level are the main limitations in obtaining an early diagnosis and implementing an earlier specific treatment [28].

There are some additional urinary markers, such as the sodium excretion rate and proteinuria, which can be useful in differentiating between a functional and morphological kidney injury; they are used only as complementary biomarkers. In these circumstances, new diagnostic tools are needed. In recent years, some new serum and urinary biomarkers have been intensively studied; these are presented in the next section [25,26,27,28,29].

### 3.1. Serum Biomarkers

Serum cystatin C (CysC) is a low molecular weight proteinase produced by all nucleated cells and is easily filtrated by the glomerulus and reabsorbed in the proximal tubular cells [30]; it is a marker for glomerular filtration function, rather than a diagnosis tool for AKI, and has been recognized by the FDA since 2002 [31,32]. In comparison to sCr, CysC is independent of sex, age, and sarcopenia [33], but it is influenced by high levels of C reactive protein, thyroid dysfunction, and smoking [7]. In addition, some studies showed that substituting cystatin C for creatinine in the MELD score significantly improved the performance of this prognostic score in predicting short-term mortality with better accuracy [34,35]; however, this needs to be further investigated in larger scale cohorts due to some controversial results in other studies [36].

### 3.2. Urinary Biomarkers

Neutrophil Gelatinase-Associated Lipocalin (NGAL) is a protein expressed by immune cells, hepatocytes, and renal tubular cells, secondary to cellular damage [37]. The proposed mechanism is that damaged renal tubular cells lead to the upregulation of NAGL expression [38]. Urine and serum NGAL are independent predictors of AKI, with a higher sensitivity and specificity of uNGAL [39]. Some older studies showed that urinary levels of NAGL are correlated with the degree of renal injury, the value being markedly increased in urinary tract infections and acute tubular necrosis (ATN) compared to those with prerenal azotemia due to renal depletion, hepatorenal syndrome, or chronic kidney disease [40]. However, more recent studies have shown that there is a significant statistical increase in serum and urinary NGAL in the early phase of AKI, as opposed to sCr [28].

Kidney injury molecule 1 (KIM-1) is an apical transmembrane glycoprotein found in proximal tubules; it is released into urine during cell injury, secondary to ischemia, as seen in ATN [41,42]. Initially, the correlation between KIM-1 and renal ischemia was proven in animal studies [43]; following that, the significant increase in this molecule in the urine was demonstrated in small human cohorts, with the confirmation of acute tubular necrosis in kidney biopsy samples [41].

Fatty acid-binding proteins (FABPs) are cytoplasmic proteins expressed in tissues with active fatty acid metabolism (initially identified in the hepatocytes and recently in the proximal tubule epithelium); they have an important role in reducing cellular oxidative stress through the binding of fatty acid oxidation products. For that reason, FABP was considered a potential marker of cell injury; this role was proven by the high urinary levels in patients with progressive AKI with poor outcomes; for these patients, renal replacement therapy is required [44]. In a study of kidney transplant recipients, the association between renal hypoxia and the increased level of FABPs was demonstrated by immunohistological analyses [45].

Interleukin-18 (Il-18) is an inflammation mediator produced by a lymphocyte T subtype (CD4) in different organs, including the renal proximal tubular cells, as a response to cellular injury. Markedly increased levels were seen in ATN [46,47]. A recent study revealed that Il-18 could be an independent risk factor for HRS in the subgroup of patients with hepatitis B-related cirrhosis [23].

Urine N-acetyl-β-D-glucosaminidase (NAG) is a large lysosomal enzyme found in proximal renal tubules. Because of its weight, it cannot penetrate the glomerulus; it is increased only during renal tubular injury and has a higher specificity for the diagnosis of ATN [48]. An elevated level of NAG is also an independent predictor of AKI development in the case of patients with decompensated cirrhosis [49].

There are some studies showing that in patients with cirrhosis and progressive AKI, the combination of multiple urinary biomarkers, including uNGAL, KIM-1, Il-18, FABPs, and NAG, has a good ability to distinguish between ATN and non-ATN and to predict the progression of renal failure and the mortality rate; however, it does not have the capacity to diagnose an early hepatorenal syndrome [6,46,50].

A short overview of the most useful biomarkers for the assessment of an AKI that is suspected to be HRS is available in Table 3. Their major limitations are related to an unspecified increase in CKD, inflammations, infections, and nephrotoxic medications.

### 3.3. Imaging Studies

Firstly, the ultrasonography of the kidney is the easiest non-invasive tool with which to exclude the morphological causes of renal damage, such as obstructive uropathy, or to raise the question of chronic kidney disease, but it cannot tell the difference between the sub-types of parenchymal renal injuries [51].

Due to well-known hemodynamic changes, such as vasoconstriction in kidney injuries related to advanced liver disease, renal Doppler ultrasonography studies were implemented to measure increased artery resistance [52]. The resistive index (RI) is a parameter used to assess vascular resistance in the small intraparenchymal arteries through the analysis of the Doppler waveform [53]. There are multiple conditions correlated with a high RI, such as acute tubular necrosis [54], obstructive renal disease [55], or renal vein thrombosis [56]. One of the leading studies which proved that there was a significant correlation between a high resistive index and the prevalence of hepatorenal syndrome was performed in 1994. Half of the cohort with a sCr level < 1.5 mg/dL and an elevated RI (cut-off > 0.7) developed renal disfunction; the threshold was more elevated (0.77 +/− 0.05) in the patients with HRS. The interlobar and the arcuate arteries were the sites of Doppler signal tracings [52].

The use of the ultrasound technique, namely point-of-care ultrasonography (POCUS) was used to confirm the diagnosis of HRS in patients with AKI. This time, the inferior vena cava diameter and (IVCD) and collapsibility index (IVCCI) were measured with the aim of assessing the volume status in AKI and guiding the need for volume expansion [57]. The data showed a great variability of IVCD and IVCCI in patients with an AKI that was suspected of being HRS; according to these data, the reclassification of the subjects in terms of depleted volume, expanded volume, and increased intraabdominal pressure requires a different approach. These results were confirmed by other studies which concluded that 40% of the patients were misdiagnosed as HRS-AKI and treated according to the measurements of IVCD and IVCCI that suggested hipo- or hypervolemia; their renal function was improved with supplementary hydration or an increase in diuretics [58].

Recently, contrast-enhanced ultrasonography (CEUS) was used in a preliminary report to evaluate the responsiveness to Terlipressin in four patients with HRS; changes in renal microcirculation were detected, as a marked increase in the perfusion index in response to vasoactive drug. The data are insufficient for validation in clinical practice as a means of predicting the response to vasoactive drugs [59].

To overcome the limitations of CEUS, which are related to subjective evaluation, an improved method named dynamic-CEUS (D-CEUS) was proposed for the evaluation of gastrointestinal neoplasic and inflammatory disorders. This technique represents a quantitative assessment of ultrasound contrast agent kinetics (UCA) in a specific region of interest (ROI), generated using time intensity curve (TIC) analysis. Two main parameters were measured: amplitude (linked to blood volume) and time (related to blood flow) [60]. D-CEUS, with the blood flow analysis of the hepatic parenchyma, was used as a non-invasive tool to predict the stage of liver fibrosis and the clinical portal hypertension in patients with compensated advanced liver disease, as seen in some of the previous studies [61,62,63]. Through knowing the patterns of HRS physiopathology, with the hemodynamic renal changes, this method could be used as an alternative non-invasive tool in the early diagnosis of HRS and should be taken into account in further studies.

Another ultrasound technique, shear wave elastography (SWE), which is used to measure tissue stiffness, was proposed in recent years as a non-invasive tool for detecting nephropathies, but with discordant results. This technique uses a real-time short-duration acoustic push pulse to generate shear waves that are propagated perpendicular to the ultrasound beam, directly to the tissue of interest, causing its deformation or displacement. The two parameters measured are the time-to-peak displacement and the recovery time, expressed in pressure units of kilopascals (kPa) and velocity (m/s), which are generated by an equation called Young’s modulus [64].

Several studies showed that cortical renal stiffness is inversely correlated with the glomerular filtration rate (eGFR), with higher values in the patients with chronic kidney disease compared to the control groups, but without well-defined cutoff values [65,66,67]. On the other hand, other studies reported a negative correlation, and the shear wave velocity decreased with the increasing stages of chronic kidney disease (CKD) [68,69]. Meanwhile, additional studies did not find any correlation linking the renal stiffness and the stage of CKD, especially in incipient stages [70,71,72].

As a consequence of multiple discrepancies between the results, in contrast to liver fibrosis, the hypothesis that renal stiffness is influenced by the perfusion pressure more than fibrosis stage was supported by a study in 2018 which compared the ultrasound shear wave elastography with magnetic resonance elastography and renal microvascular flow in the assessment of CKD in transplanted patients [73].

Initially, this bond between the variation in renal stiffness and the hemodynamic changes in the kidneys was proved by two animal studies [74,75]. In 2012, Gennisson et al. discovered that renal artery ligation in three pigs led to a decrease in renal stiffness, as opposed to renal vein ligation, resulting in vascular congestion and, consequently, an increase in renal stiffness [73].

Furthermore, in another porcine magnetic resonance elastography (MRE) model, Castelein et al. demonstrated that kidney stiffness—especially if measured in the renal cortex—is strongly and directly correlated with both renal blood flow and systemic arterial pressure (r = 0.91 and 0.96, respectively, *p* < 0.001 in both cases) [76]. These data led to the conception of a theory of renal softening as secondary to reduced perfusion in kidney injury [77]. In the context of liver cirrhosis, these data were confirmed in a prospective analysis of a small cohort of 21 cirrhotic patients with ascites, showing that in patients with hepatorenal syndrome, the renal stiffness—measured using an MRE at 90 Hz—was significantly lower compared to that of the patients with normal renal function (3.3 kPa vs. 5.08 kPa; *p* ≤ 0.014; AUROC = 0.94) [78]. The link between the decrease in renal stiffness and ischemic kidney injury is summarized in Figure 3.

However, there is a single study from 2021 concerning the prognostic value of renal stiffness assessed by ultrasound shear wave elastography (SWE) in predicting the 30-day development of hepatorenal syndrome. The authors found that decreased baseline renal stiffness can predict kidney injury in patients with advanced liver disease and ascites alongside increased serum cystatin C, urinary NGAL, and RARI. In this study, the combination of SWE and the renal artery resistive index improved the ability to predict the occurrence of HRS, with an AUROC of 0.89 [79].

## 4. Concluding Remarks and Future Perspectives

There is a permanent need to improve the diagnosis criteria due to the multiple limitations of serum creatinine and the delay in the diagnosis of hepatorenal syndrome. Consequently, even if there are favorable results for the new biomarkers (especially NGAL and sCys), further studies are needed with larger cohorts for broader validation. The use of POCUS also has to be reinforced, as this is easy to use and available, and it can reclassify AKI and modify the initial management of these patients. Non-invasive ultrasound-based imaging studies, such as shear wave elastography assessment associated with resistive index measurement or contrast-enhanced ultrasound, could represent a better, disease-specific, repeatable, easy-to-perform, and cost-efficient technique to predict renal tissue damage or repair in patients with advanced liver disease and AKI. However, large, multicentric validation studies are warranted to further validate these techniques in this clinical scenario.

## Figures and Tables

**Figure 1 diagnostics-14-00938-f001:**
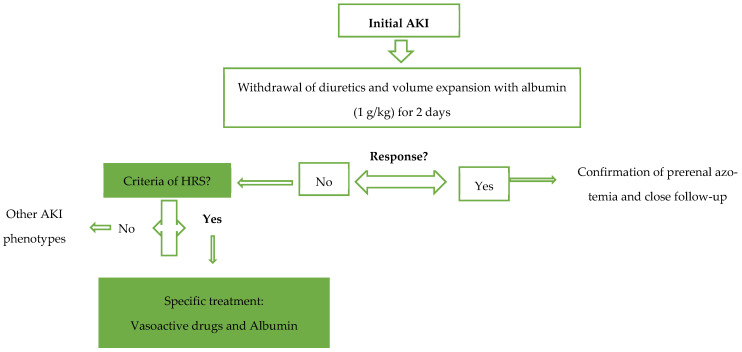
Algorithm for the management of AKI in patients with cirrhosis.

**Figure 2 diagnostics-14-00938-f002:**
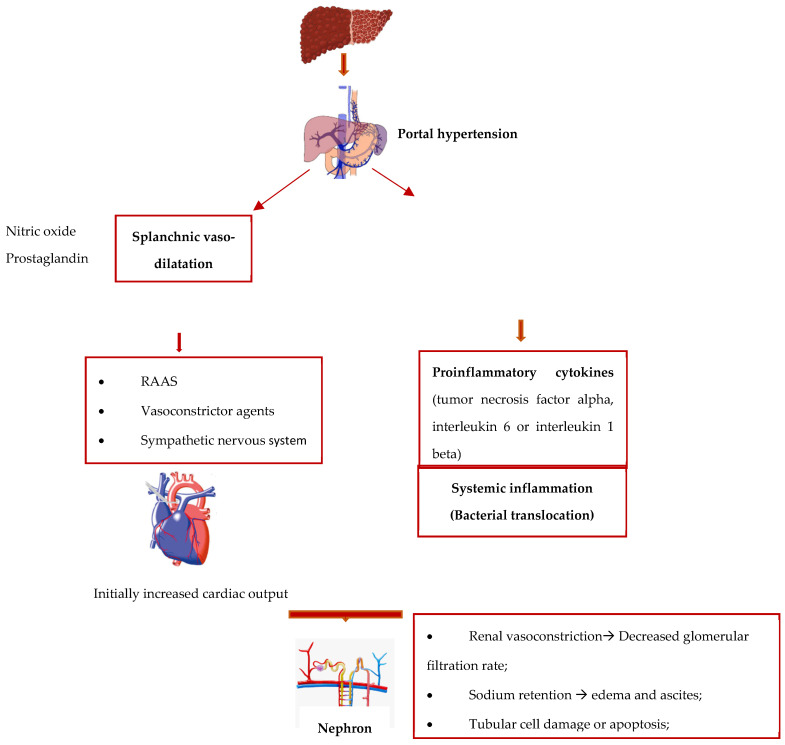
Physiopathology of HRS.

**Figure 3 diagnostics-14-00938-f003:**
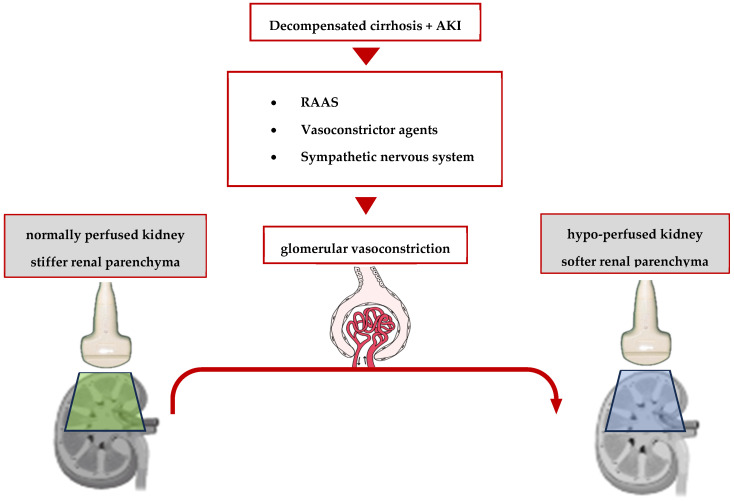
Renal hemodynamics in cirrhotic patients with AKI.

**Table 1 diagnostics-14-00938-t001:** The proposed classification of renal dysfunction in patients with liver cirrhosis by the Acute Dialysis Quality Initiative and the International Club of Ascites work group.

Classification of Kidney Injury in Patients with Cirrhosis	
Acute kidney injury (AKI) Increase in sCr of ≥50% from baseline or an increase in sCr of ≥0.3 mg/dL in <48 h.	Stage 1: Increase in sCr ≥ 0.3 mg/dL or an increase in sCr 1.5-fold to 2-fold from baseline Stage 2: Increase in sCr > 2-fold to 3-fold from baseline Stage 3: Increase in sCr > 3-fold from baseline or an increase in sCr ≥ 4.0 mg/dL or initiation of renal replacement therapy
Chronic kidney disease (CKD)	Glomerular filtration rate (GFR) of <60 mL/min for >3 months (MDRD6 formula) HRS type 2 is a specific form of CKD
Acute-on-chronic kidney disease	Increase in sCr of ≥50% from baseline or an increase in sCr of ≥0.3 mg/dL in <48 h in a patient with cirrhosis whose GFR is <60 mL/min for >3 months (MDRD6 formula)

**Table 2 diagnostics-14-00938-t002:** The International Club of Ascites diagnostic criteria for hepatorenal syndrome.

International Club of Ascites Diagnostic Criteria for Hepatorenal Syndrome
Diagnosis of cirrhosis and ascites Diagnosis of acute kidney injury (AKI) according to ICA-AKI criteria No response after 48 h of diuretic withdrawal and plasma volume expansion (not necessarily with albumin at 1 g per kg of body weight) Absence of shock No current or recent use of nephrotoxic agents No signs of structural kidney injuries, defined as the following: ● Absence of proteinuria (>500 mg per day or equivalent) ● Absence of microscopic hematuria (>50 red blood cells per high-power field) ● Normal findings on renal ultrasonography

**Table 3 diagnostics-14-00938-t003:** An overview of the most relevant biomarkers studied in HRS-AKI.

Biomarker	Place of Origin	Fluid Tested	Time of Expression	Clinical Relevance
CysC [30,34,35]	Nucleated cells	Serum	12–24 h	Correlation with renal function, 5-year survival, 1-year AKI development, 3-month mortality, HRS development
NGAL [39,40]	Loop of Henle and collecting ducts	Urine/serum	<12 h	AKI phenotype (differentiation of ATN), HRS development, 3-month mortality, AKI progression
KIM-1 [41,43]	Proximal tubular cells	Urine	<12 h	Discriminating ATN
FABPs [44,45]	Proximal tubular cells, hepatocytes	Urine	<12 h	Discriminating ATN
IL-18 [23,46,47]	Monocytes, macrophages, dendritic cells	Urine	<12 h	Discriminating ATN, 3-month mortality
NAG [48,49]	Proximal tubular cells	Urine	12 h	Discriminating HRS-AKI, predicting the 3-month transplant-free survival

## Data Availability

Not applicable.

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
