# Peer review of "Seeing and Sensing the Hepatorenal Syndrome (HRS): The Growing Role of Ultrasound-Based Techniques as Non-Invasive Tools for the Diagnosis of HRS"

_diagnostics, 2024, doi:10.3390/diagnostics14090938_

Round 1
Reviewer 1 Report
Comments and Suggestions for Authors
Taluta el al. have written a review of non-invasive tools for the diagnosis of hepatorenal syndrome. Based on the title, I though the focus would just be on imaging but they include biomarkers as well. It would be useful if more of the paper was focused on these tools rather than background about HRS. There is plenty of room to expand the discussion of biomarkers, little detail is given for them. The authors could consider a table reviewing the prominent biomarkers papers in patients with HRS. This would be more practically useful for the paper than those with the diagnostic criteria for AKI and HRS.
As far as imaging modalities, there is a very long section about shear wave elastography but only at the end does it touch on its use in the topic of the paper. There are some other papers, particularly the one by JCQ Velez, looking at POCUS and HRS diagnosis, it would be good to include discussion of these, especially as this is a technique now in common usage unlike SWE or CEUS.
In section 2.1, in the first paragraph, it seems to state that HRS occurs in almost 50% of hospitalized patients, which certainly in not the case. Do the authors mean AKI occurs in nearly 50% of hospitalized patients with cirrhosis? In the 3rd paragraph, it states that vasopressin is the first choice vasoconstrictor for HRS. This is incorrect, the first line choice is terlipressin, with norepinephrine first-line when terlipressin is not available. Vasopressin is very rarely used for HRS.
Comments on the Quality of English LanguageThe manuscript could use some editing for English grammar and sentence construction.
Author Response
see the document attached

Reviewer 2 Report
Comments and Suggestions for Authors
This manuscript is a review focused on diagnosis of renal disorders in patients with decompensated liver cirrhosis. This topic is important due to frequent change of diagnostic criteria of acute kidney injury in liver cirrhosis and definition of hepato-renal syndrome (HRS) resulting from lack of sensitive markers of renal ischemia.
In this paper I have found several claims that are not in accordance with pathophysiological knowledge on portal hypertension.
In section “The Spectrum of Acute Kidney Injuries in Cirrhosis” the Authors itemized 3 main phenotypes of AKI in liver cirrhosis, namely prerenal, intrarenal – mainly represented by acute tubular necrosis and post-renal AKI (obstructive uropathies). In this division there is no place for HRS, which is neither prerenal AKI responding to plasma volume repletion nor acute tubular necrosis.
In section “The Hepatorenal Syndrome” beta-blockers are named “vasodilators”. To my knowledge non-selective beta-blockers are peripheral vasoconstrictors and mesenteric vasoconstriction is one of major mechanisms of their portal hypotensive effect.
I would also disagree that Vasopressin is first choice drug in treatment of HRS. Terlipressin due to longer half-time and milder side-effects is believed to be a better choice.
In this section there is phrase “The mechanism of vasoconstrictive drugs is to counteract the hypovolemia due to splanchnic arterial vasodilation by improving renal perfusion”. Generally, patients with decompensated cirrhosis are hypervolemic with altered distribution of plasma volume leading to effective hypovolemia. The paradox that vasoconstrictors improve perfusion in vasoconstricted kidneys is not well explained.
In phrase „a complete response is defined as a sCr within 0.3 mg/dl” something is missing.
In section “Pathophysiology of Hepatorenal Syndrome” it has been said that splanchnic arterial vasodilation is secondary to reduced systemic vascular resistance. It is opposite, reduced systemic vascular resistance is secondary to splanchnic arterial vasodilation. In this context, what is compensated by increased cardiac output?
The phrase “Secondary to hypovolemia, the vasoconstriction in the renal arterial vessels will respond with the activation of the systemic vasoconstrictor mechanisms (the renin-angiotensin-aldosterone system, rising the production of vasoconstrictor agents” is not clear. Besides “hypovolemia” should be replaced by “effective hypovolemia” that denotes underfilled central circulation.
Other claim is that “hyponatremia is due to solute-free water elimination”. Rather failure of solute-free water elimination.
In section “Diagnosis Tools” information on necessity to withdraw diuretics and administration of albumin has been repeated from previous pages. Besides, naming diuretics nephrotoxic drugs is somewhat exaggerated.
Author Response
see the document attached

Reviewer 3 Report
Comments and Suggestions for Authors
This article nicely reviews the current practices in non-invasive tools in the diagnosis of hepatorenal syndrome. My main concern is the substandard of English language that must be polished as well as proper use of terminologies and abbreviations. Please see the attached manuscript file for detailed remarks.

English language must be reviewed thoroughly.
Author Response
see the document attached

Round 2
Reviewer 2 Report
Comments and Suggestions for Authors
None